# METTL14-upregulated miR-6858 triggers cell apoptosis in keratinocytes of oral lichen planus through decreasing GSDMC

Xiangyu Wang[1,2,3,6], Shuangting Li[1,6], Huimin Song[1,2,3,6], Yan Ding[4,5,6], Ruifang Gao[1], Xiaotong Shi[1,2,3], Ran Li[1] & Xuejun Ge [1✉]

Oral lichen planus (OLP), a chronic inflammatory disorder, is characterized by the massive cell apoptosis in the keratinocytes of oral mucosa. However, the mechanism responsible for triggering oral keratinocyte apoptosis is not fully explained. Here, we identify that Gasdermin C (GSDMC) downregulation contributes to apoptosis in human oral keratinocytes. Mechanistically, we describe that activated nuclear factor kappa B (NF-κB) pathway induces overexpression of methyltransferase-like 14 (METTL14), which increases $N^6$-adenosine methylation (m6A) levels in the epithelial layer of OLP. m6A modification is capable of regulating primary miR-6858 processing and alternative splicing, leading to miR-6858 increases. miR-6858 can bind and promote *GSDMC* mRNA degradation. Forced expression of GSDMC is able to rescue cell apoptosis in human oral keratinocyte models resembling OLP. Collectively, our data unveil that m6A modification regulates miR-6858 production to decrease GSDMC expression and to trigger keratinocyte apoptosis in the context of OLP.

[1] Shanxi Province Key Laboratory of Oral Diseases Prevention and New Materials, Shanxi Medical University School and Hospital of Stomatology, Taiyuan, Shanxi, China. [2] Department of Child Dental and Preventive Dentistry, Shanxi Medical University School and Hospital of Stomatology, Taiyuan, Shanxi, China. [3] Department of Oral Medicine, Shanxi Medical University School and Hospital of Stomatology, Taiyuan, Shanxi, China. [4] Department of Dermatology, Hainan Provincial Hospital of Skin Disease, Haikou, Hainan, China. [5] Department of Dermatology, Skin Disease Hospital of Hainan Medical University, Haikou, Hainan, China. [6]These authors contributed equally: Xiangyu Wang, Shuangting Li, Huimin Song, Yan Ding. ✉email: gxj19722003@163.com

O ral lichen planus (OLP) is considered to be a chronic inflammation-related disorder affecting oral mucosa[1]. The range of OLP prevalence is between 0.22% and 5% across the world[2]. OLP patients often suffer from these clinical symptoms such as burning, soreness and pain[3,4]. These symptoms become worsened amid chewing or tooth brushing, limiting patients' daily activities[3,4]. Albeit the pathogenesis of OLP remains elusive, autoimmune response, genetic factors, infectious agents and psychological factors all contribute to the onset and development of OLP[4]. To date, OLP is identified to be initiated by basal keratinocytes apoptosis triggered by T lymphocytes in the oral stratified squamous epithelium[5–7]. Exploring the cell apoptosis process of oral keratinocytes might be an efficient strategy for OLP management.

Post-transcriptional modifications of RNAs are regarded as an indispensable mechanism regulating gene expression[8]. In mammalian cells, $N^6$-adenosine methylation (m6A) of polyadenylated RNAs is considered to be the most prevalent internal modification in mRNAs at the post-transcriptional level[9]. The adenosine nucleotides regulated by m6A methylation in mammalian RNAs are approximately 0.4%, with 3–5 m6A modifications on average in each mRNA[10,11]. The deposition of m6A modification is organized in the mammalian cells' transcriptome. The m6A sites have a consensus motif RRm6ACH [(G/A/U)(G > A) m6AC(U > A > C)], with enrichment in the coding sequence (CDS) and 3'untranslated region (3'UTR)[12–14]. RNA m6A modification is regulated by writers, erasers and readers dynamically to achieve a reversible balance. Writers are methyltransferases responsible for installing the methyl group on the $N^6$ position of adenosine, while erasers are demethylases which remove methyl group from it reversibly. Readers are proteins accounting for interacting with and recognizing the m6A modification sites[15]. The methyltransferase of m6A modification is a complex consisting of Wilms tumor 1-associated protein (WTAP), methyltransferase-like 3 (METTL3) and METTL14. AlkB homolog 5 (ALKBH5) and fat mass and obesity-associated protein (FTO) are the two m6A erasers identified so far[10,11]. Three groups of m6A readers, YT521-B homology (YTH) domain-containing proteins (YTHDF1/2/3 and YTHDC1/2), heterogeneous nuclear ribonucleoproteins (HNRNPA2B1 and HNRNPC) and insulin-like growth factor 2 mRNA-binding proteins (IGF2BP1/2/3), utilizes RNA binding domains and RNA recognition motif to determine the fate of the modified mRNAs[15,16]. Although m6A modification is identified to regulate tumor growth, inflammatory responses and stem cell pluripotency[17–19], its roles in OLP initiation and development remain unanswered.

Gasdermin (GSDM) family includes GSDMA/B/C/D/E and Pejvakin, which are highly expressed in keratinocytes and intestinal epithelial cells[20,21]. The main role of GSDM family is identified to enhance pyroptotic cell death[22–24]. However, recent studies have suggested that GSDMC in cancer cells is capable of switching apoptosis to pyroptosis, thereby leading to tumor necrosis[25]. Given that cell apoptosis is a typical feature of OLP, we set to investigate the functions of GSDMC in oral keratinocytes in the context of OLP. In this study, our results reveal that up-regulation of METTL14 in the setting of OLP promotes m6A methylation of primiR-6858, which results in miR-6858 over-expression. Increased miR-6858 targets and decreases *GSDMC* mRNA to induce cell apoptosis in oral keratinocytes.

## Results

### GSDMC expression is decreased in the keratinocytes of OLP.

Since GSDMC expression in the epithelium of OLP is not clear, we detected the levels of it by real-time PCR and western blot. As shown in Fig. 1, both mRNA and protein levels of GSDMC were decreased largely in the epithelia derived from OLP biopsies compared to those from healthy controls (Fig. 1a–c). To establish cell models mimicking OLP in vitro, we used both lipopolysaccharides (LPS) and the supernatants of culture medium from activated $CD4^+$ T cells to challenge human oral keratinocytes (HOKs)[26,27]. Consistently, GSDMC expression showed a

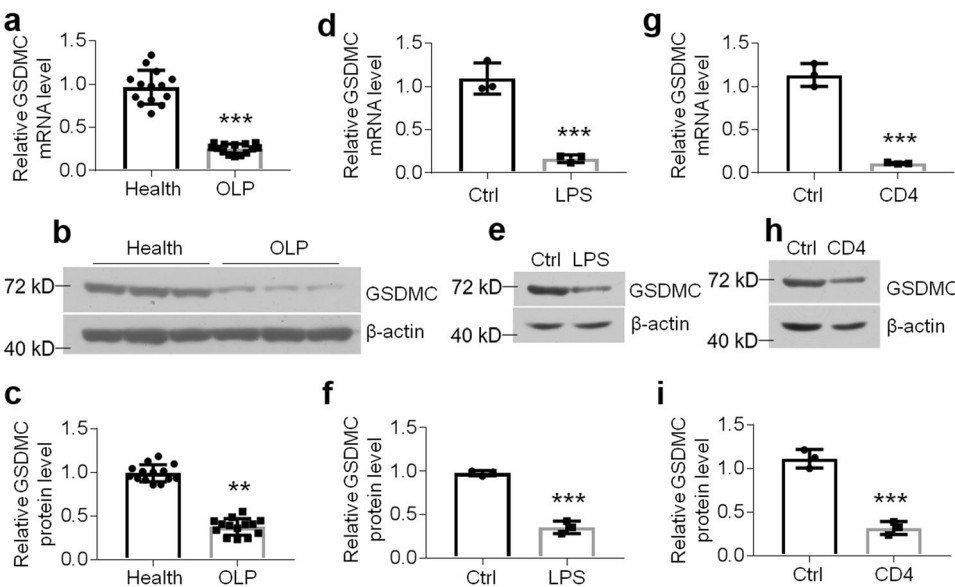

**Fig. 1 GSDMC expression is decreased in OLP. a** *GSDMC* mRNA levels in oral epithelial layers from healthy or diseased tissues determined by real-time PCR, n = 14. Western blot (**b**) and densitometric analysis (**c**) showing GSDMC protein levels in oral epithelial layers from healthy or diseased tissues, n = 14. Data (**b**) are representative of 14 samples. GSDMC mRNA or protein expression detected by real-time PCR (**d**), western blot (**e**) or densitometric quantitation (**f**) in HOKs with or without 12-hour LPS treatment (100 ng/ml), n = 3. Data (**e**) are representative of 3 samples. GSDMC mRNA or protein levels measured by real-time PCR (**g**), western blot (**h**) or densitometric quantitation (**i**) in HOKs with or without 12-hour activated $CD4^+$ T cells treatment, n = 3. Data (**h**) are representative of 3 samples. **\*\***P < 0.01, **\*\*\***P < 0.001 vs corresponding health or control group; Ctrl, control. Data were expressed as means ± standard deviation. All experiments were carried out at least 3 times. Student's *t* test (**a**, **c**, **d**, **f**, **g** and **i**) was performed for the statistical analysis.

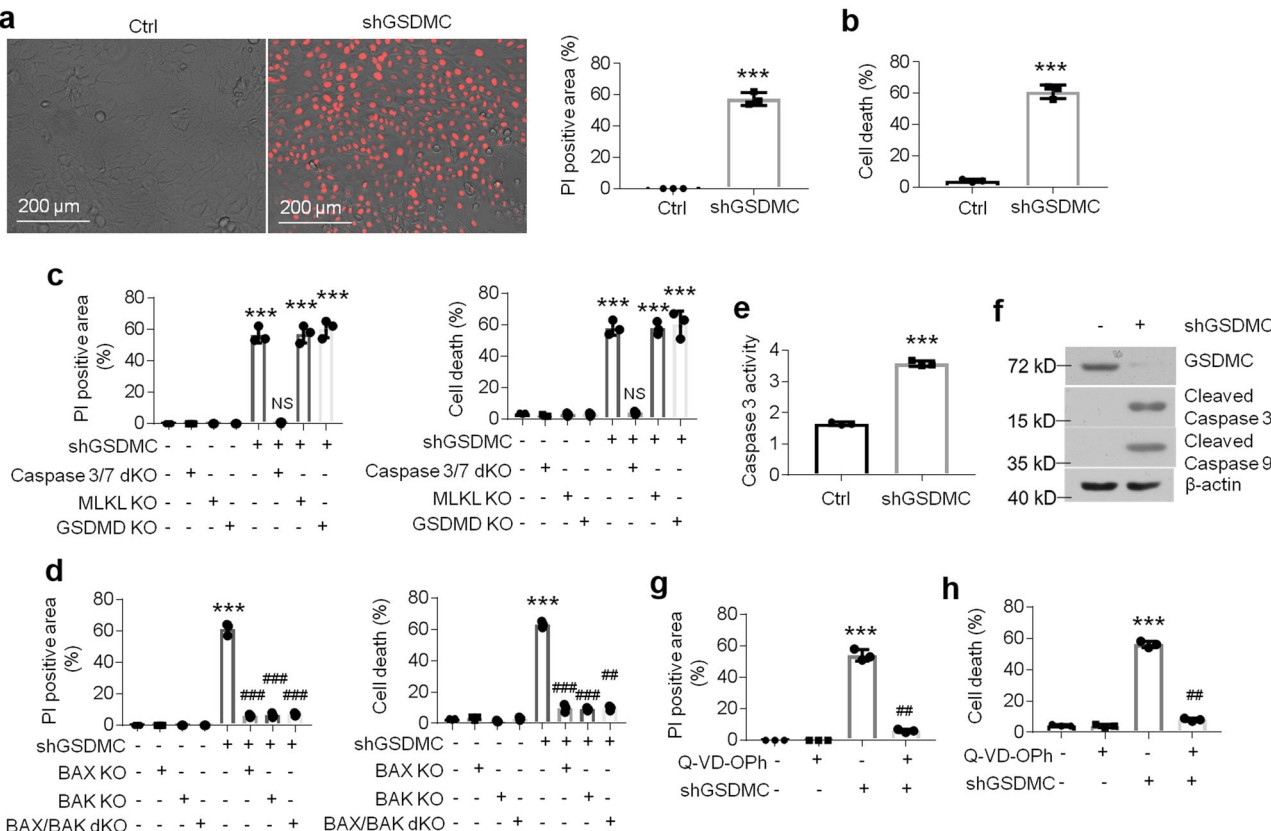

**Fig. 2 GSDMC regulates apoptosis in oral keratinocytes. a** PI staining and quantitative analysis in HOKs transduced with control- or shGSDMC-lentivirus. **b** Cell death detection of HOKs transduced with control- or shGSDMC-lentivirus. **c** PI staining analysis and cell death measurement of HOKs with GSDMC knockdown and caspase3/7, MLKL or GSDMD knockout. **d** PI staining analysis and cell death measurement of HOKs with GSDMC knockdown and BAX or BAK knockout. Caspase 3 activity and proteins levels in HOKs transduced with control- or shGSDMC-lentivirus evaluated by caspase 3 activity assay (**e**) or western blot (**f**). PI staining analysis (**g**) and cell death measurement (**h**) of HOKs transduced with control- or shGSDMC-lentivirus. HOKs were pretreated with or without Q-VD-OPh (100 nM) for 4 h. ***$P < 0.001$ vs corresponding control group; ##$P < 0.01$, ###$P < 0.001$ vs corresponding LPS or CD4 group; $n = 3$; Ctrl, control; KO, knockout; dKO, double knockout. Data were expressed as means ± standard deviation. All experiments were carried out at least 3 times. Student's $t$ test (**a**, **b**, **e**) and one-way ANOVA (**c**, **d**, **g**, **h**) were used for statistical analysis.

remarkable decrease in the two cell models at both mRNA and protein levels (Fig. 1d–i). Importantly, GSDMC expression was down-regulated in a time course-dependent manner upon LPS or activated CD4[+] T cells treatment (Supplementary Fig. 1), indicating that these stimulators decreased GSDMC levels in oral keratinocytes.

**Down-regulation of GSDMC induces cell apoptosis in oral keratinocytes.** As previous studies suggested that GSDMC plays a critical role in cell apoptosis in tumor cells[25], we investigated the effects of GSDMC on cell death in HOKs. The propidium iodide (PI) staining has been used for the assessment of cell apoptosis in different experimental models widely[28]. PI is able to bind to double stranded DNA in dead cells with defective plasma membranes, but is excluded from alive cells[28]. After binding with DNA, the quantum yield of PI is increased 20-30 fold and can be detected by fluorescence microscope[28,29]. As exhibited in Fig. 2, PI staining data demonstrated that most of oral keratinocytes were dead upon GSDMC knockdown (Fig. 2a). Cell death of HOKs was further confirmed by enzymatic assays (Fig. 2b). To explain the manner of cell death induced by GSDMC deficiency, we generated Caspase3/7 double knockout (dKO), Gasdermin D (GSDMD) KO, Mixed Lineage Kinase Domain Like Pseudokinase (MLKL) KO, BAX KO or BAK KO HOKs to investigate the roles of apoptosis, pyroptosis and necroptosis in HOKs' death[30] (Supplementary Fig. 2a). Of note, blockage of apoptosis pathway

decreased HOKs' death upon the loss of GSDMC (Fig. 2c, d). Moreover, caspase 3 activity, cleaved caspase 3 and cleaved caspase 9 levels were all elevated in HOKs after shGSDMC-lentivirus infection (Fig. 2e, f), noting that GSDMC knockdown induced cell apoptosis in HOKs. The apoptosis inhibitor, Q-VD-OPh, rescued deficient GSDMC-induced cell death (Fig. 2g, h). In comparison to our previous data[27,31], we found that GSDMC expression showed positive correlations with the levels of anti-apoptotic factors such as vitamin D receptor (VDR), miR-26a/b, B-cell lymphoma-2 (BCL-2) in the epithelial layers of human biopsies, but had negative correlations with miR-802 and cleaved caspase 3 which are known as pro-apoptotic factors (Supplementary Fig. 2b). Importantly, mitochondrial membrane potential was changed after GSDMC knockdown (Supplementary Fig. 3a, b), indicating the mitochondrial dysfunction. Overall, these findings uncover that GSDMC plays a protective role in oral keratinocytes in the setting of OLP. The excessive cell apoptosis in the epithelial layer of OLP mucosa has been identified in our previous studies[31].

**GSDMC protects HOKs from apoptosis in the cell models of OLP.** To test whether GSDMC overexpression is able to block cell apoptosis, we treated HOKs with GSDMC-expressing lentivirus in the presence of LPS or activated CD4[+] T cells. Accordingly, forced expression of GSDMC relieved cell death and pro-apoptotic pathway activation induced by challenges (Fig. 3a–h

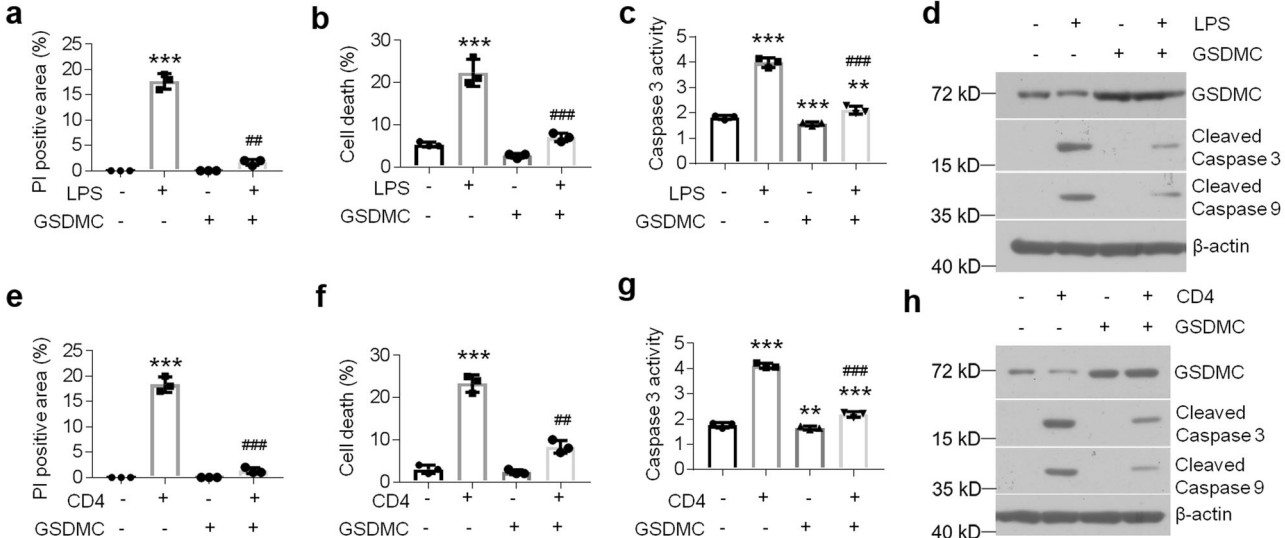

**Fig. 3 GSDMC inhibits apoptosis in oral keratinocytes.** PI staining analysis (**a**), cell death measurement (**b**), caspase 3 activity assay (**c**) or western blot (**d**) in control- or GSDMC-lentivirus-transduced HOKs with or without 12-hour LPS challenge (100 ng/ml). PI staining analysis (**e**), cell death measurement (**f**), caspase 3 activity assay (**g**) or western blot (**h**) in control- or GSDMC-lentivirus-transduced HOKs with or without 12-hour activated CD4$^+$ T cells stimulation (30% final volumetric concentration). **$P < 0.01$, ***$P < 0.001$ vs corresponding control group; ##$P < 0.01$, ###$P < 0.001$ vs corresponding LPS or CD4 group; $n = 3$; Data were expressed as means ± standard deviation. All experiments were carried out at least 3 times. One-way ANOVA was used for statistical analysis.

and Supplementary Fig. 3c, d). Interestingly, the overexpression of GSDMC in HOKs did not lead to cell death (Fig. 3a, b), indicating GSDMC, at least itself alone, could not trigger pyroptosis in HOKs.

**m$^6$A methylation levels are enhanced in the epithelial layer of OLP tissues due to the boosted *METTL14* transcripts.** m$^6$A modification is reported to be a prevalent way to regulate gene expression[10]. To determine whether GSDMC expression was mediated by m$^6$A methylation in oral keratinocytes, we detected m$^6$A status in the epithelia of human specimens. Of interest, m$^6$A modification levels showed a robust increase in the epithelial layer of OLP samples compared to that of controls (Fig. 4a), and also in the HOKs with treatments as indicated (Supplementary Fig. 4a, b). To explore the underpinning by which m$^6$A is increased, we measured the writers and erasers expression in the oral keratinocytes of human samples. As shown, only *METTL14* levels were changed considerably in the diseased samples among all of the methyltransferases and demethylases (Fig. 4b and Supplementary Fig. 4c–f). METTL14 protein levels were also increased in the diseased samples (Fig. 4c, d). In accordance, METTL14 expression was up-regulated in HOKs upon LPS or activated CD4$^+$ T cells treatment (Supplementary Fig. 4g–j). Forced expression of METTL14 in HOKs was capable of elevating m$^6$A levels (Fig. 4e, f). Next, to search the underpinning of METTL14's upregulation, we screened the promoter region of human *METTL14* and found a putative nuclear factor-κB (NF-κB) binding site (Fig. 4g). Chromatin immunoprecipitation (ChIP) data showed that NF-κB p65 selectively bound with this putative κB site in HOKs using a primer flanking the potential motif (Fig. 4h). Overexpression of IκB kinase β (IKKβ) which can activate NF-κB pathway improved METTL14 levels in HOKs (Fig. 4i, j). These results suggest that m$^6$A methylation is increased in oral keratinocytes owing to METTL14 upregulation in the context of OLP. The activated NF-κB pathway in the epithelial layer of OLP biopsies has been confirmed in our previous studies[32].

To interrogate whether the *GSDMC* mRNA in HOKs is regulated by RNA m$^6$A methylation, we mined the online m$^6$A-

IP-sequencing database (GSE213714). After analysis, we found 4 putative m$^6$A sites in the *GSDMC* mRNA from human oral keratinocytes (Supplementary Fig. 4k). However, the levels of m$^6$A-related RNA fragments were comparable in IgG control and m$^6$A groups (Supplementary Fig. 4l), indicating m$^6$A methylation did not directly contribute to *GSDMC* mRNA decreases in HOKs.

**miR-6858 targets and degrades *GSDMC* mRNA in HOKs.** miRNAs are known to regulate gene expression in mammalian cells[33]. Here, targetscan database (http://www.targetscan.org/vert_72/) was used to predict the potential miRNAs responsible for *GSDMC* mRNA decreases. Based on the target score, we selected the top 5 miRNAs closely related to human *GSDMC*. Among them (miR-4795-3p, miR-548ax, miR-548ao-5p, miR-4689, miR-6858), only miR-6858 levels were boosted in the diseased human biopsies as well as the serum and saliva derived from OLP patients compared to those from healthy participants (Fig. 5a and Supplementary Fig. 5a–d). In line with these data relative to human samples, LPS or activated CD4$^+$ T cells could induce miR-6858 levels in HOKs as well (Supplementary Fig. 5e, f). The miR-6858-targeted site in human *GSDMC* mRNA was exhibited in Fig. 5b. To verify the regulatory role of miR-6858 in *GSDMC* mRNA stability, we inserted the fragment containing the miR-6858-targeted site in the 3'UTR of *GSDMC* mRNA for luciferase assays. As shown, luciferase activity of HOKs transfected with pGL3-GSDMC carrying the miR-6858-targeted site, but not that with pGL3-GSDMCmut, was suppressed by forced expression of miR-6858 (Fig. 5c), confirming miR-6858 promoted *GSDMC* mRNA degradation. Next we applied gain-of-function and loss-of-function assays to detect the effects of miR-6858 on GSDMC expression. Accordingly, miR-6858 overexpression down-regulated GSDMC levels largely in HOKs at both mRNA and protein levels which were confirmed by real-time PCR and western blot assays (Fig. 5d, e and Supplementary Fig. 5g, h). On the contrary, miR-6858 inhibitors reversed LPS- or activated CD4$^+$ T cells-induced GSDMC decreases in HOKs (Fig. 5f–i and Supplementary Fig. 5i–k).

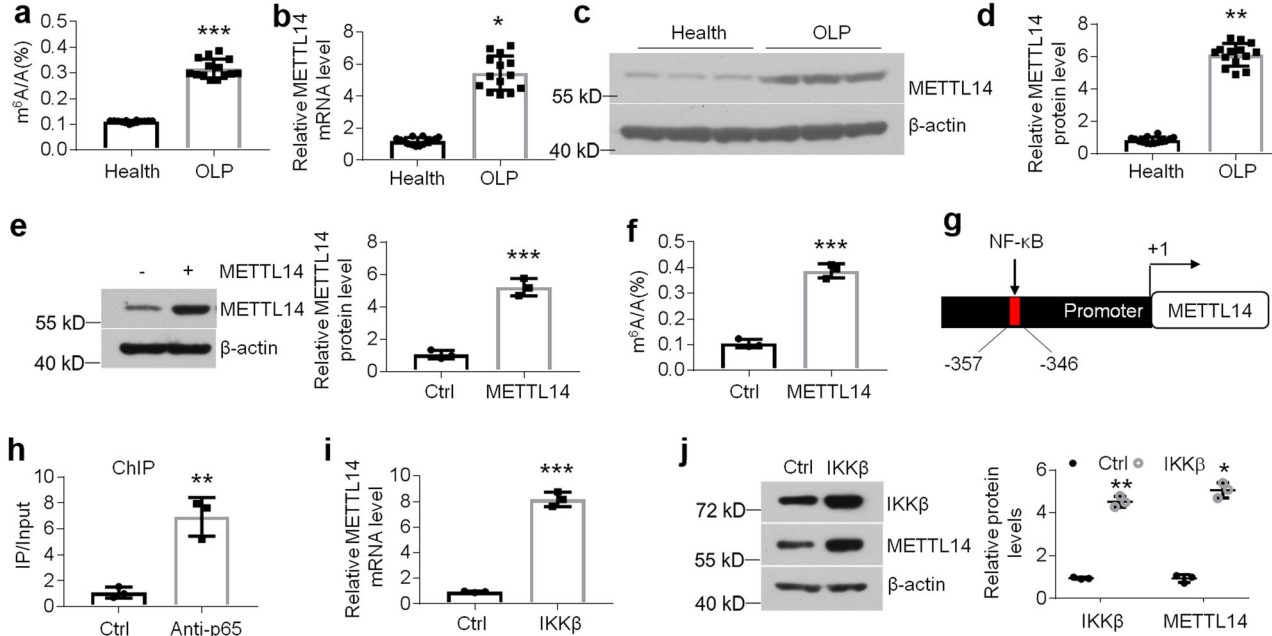

**Fig. 4 m6A modification and METTL14 levels in oral keratinocytes. a** Quantitation of m6A level in total RNAs isolated from oral keratinocytes of human mucosa, $n = 14$. METTL114 mRNA or protein expression assessed by real-time PCR (**b**), western blot (**c**) or densitometric quantitation (**d**) in oral keratinocytes of human samples, $n = 14$. METTL14 or m6A levels in HOKs transduced with control- or METTL14-expressing lentivirus determined by western blot (**e**) or m6A quantification assay (**f**), $n = 3$. **g** Schematic showing the potential NF-κB motif in the promoter of human *METTL14* gene. **h** ChIP assays of HOKs against control or anti-p65 antibody, $n = 3$. METTL14 mRNA or protein expression in HOKs with control- or IKKβ-expressing lentivirus determined by real-time PCR (**i**) or western blot (**j**), $n = 3$. *$P < 0.05$, **$P < 0.01$, ***$P < 0.001$ vs corresponding control or health group; Ctrl, control. Data were expressed as means ± standard deviation. All experiments were carried out at least 3 times. Student's *t* test (**a**, **b**, **d**, **e**, **f**, **h**, **i** and **j**) was used for statistical analysis.

**m6A methylation of primiR-6858 modulates the processing of miR-6858 in HOKs.** To uncover the mechanism by which miR-6858 is upregulated in OLP, we screened the sequence of primiR-6858 using SRAMP database and found two m6A motifs, one is within premiR-6858 sequence and the other one is not (Supplementary Fig. 6a). we then designed primers flanking the m6A motif (GGACA) in the primiR-6858, but not in the premiR-6858, to perform m6A immunoprecipitation (IP)-quantitative PCR (qPCR). As shown in Fig. 5S, m6A antibody highly bound with this m6A site in HOKs (Supplementary Fig. 6b). Next, we used single-stranded methylated RNA bait (ss-m6A) or unmethylated control RNA (ss-A) to perform RNA pull-down assay (Supplementary Fi. 6c). m6A writers, METTL14 and METTL3, were confirmed to interact with the methylated RNA bait selectively with a higher affinity compared to the unmethylated control group (Supplementary Fig. 6d).

m6A methylation has an effect on primary miRNA processing and splicing[34]. We next overexpressed or knocked down METTL14 or METTL3 levels in HOKs to observe primiR-6858 and miR-6858 expression separately (Fig. 4e and Supplementary Fig. 6e–h). Forced expression of METTL14 or METTL3 enhanced miR-6858 level and inhibited primiR-6858 expression in HOKs (Fig. 6a, b), while knockdown of m6A writer played a reversed role in them (Fig. 6c, d). A R298P mutant METTL14 and a D394A/W397A METTL3 mutant are enzymatically inactive and defective in regulating RNA m6A modification[35]. In this investigation, mutant METTL14 or METTL3 did not change or rescue primiR-6858 and miR-6858 levels as wild type METTL14 or METTL3 did (Fig. 6a–d), suggesting the modulation of primiR-6858 was dependent on RNA m6A methylation. Overexpression of FTO, a demethylase, had the opposite functions in miRNAs compared to METTL14 or METTL3 (Fig. 6e and Supplementary Fig. 6i). HNRNPA2B1 is reported to be a reader

protein in the m6A modification of primary miRNAs[34]. Similar to previous studies[34], we found HNRNPA2B1 could bind with m6A site in the primiR-6858 in HOKs and regulate miR-6858 expression (Supplementary Fig. 6j, k and Fig. 6f).

**Inhibition of miR-6858 or METTL14 rescues cell apoptosis in oral keratinocytes.** As METTL14-induced miR-6858 overexpression can promote *GSDMC* mRNA degradation to trigger cell apoptosis in HOKs, we next sought to determine whether METTL14 or miR-6858 inhibition can attenuate cell apoptosis under OLP conditions. As shown, LPS or activated CD4+ T cells-induced cell apoptosis was ameliorated in METTL14-knockdown cells (Fig. 7a, b), and so was it in cells with miR-6858 inhibition (Fig. 7c, d).

**GSDMC overexpression rescues METTL14- or miR-6858-induced cell apoptosis.** We further evaluated whether over-expression of METTL14 or miR-6858 could induce cell apoptosis in HOKs as LPS or activated CD4+ T cells did. Intriguingly, either METTL14- or miR-6858 overexpression triggered cell death and pro-apoptotic factors production in HOKs (Fig. 8a–e). However, forced expression of GSDMC rescued METTL14- or miR-6858-induced cell apoptosis (Fig. 8a–e), indicating METTL14- or miR-6858 initiated cell apoptosis in a GSDMC-dependent manner.

## Discussion

Unlike other members of GSDM family (GSDMD and GSDME), the function of GSDMC in mammalian cells is still not fully explained[36]. In this study, we confirmed that GSDMC expression shows a decrease in the keratinocytes of OLP and the down-regulation of GSDMC in oral keratinocytes results in cell

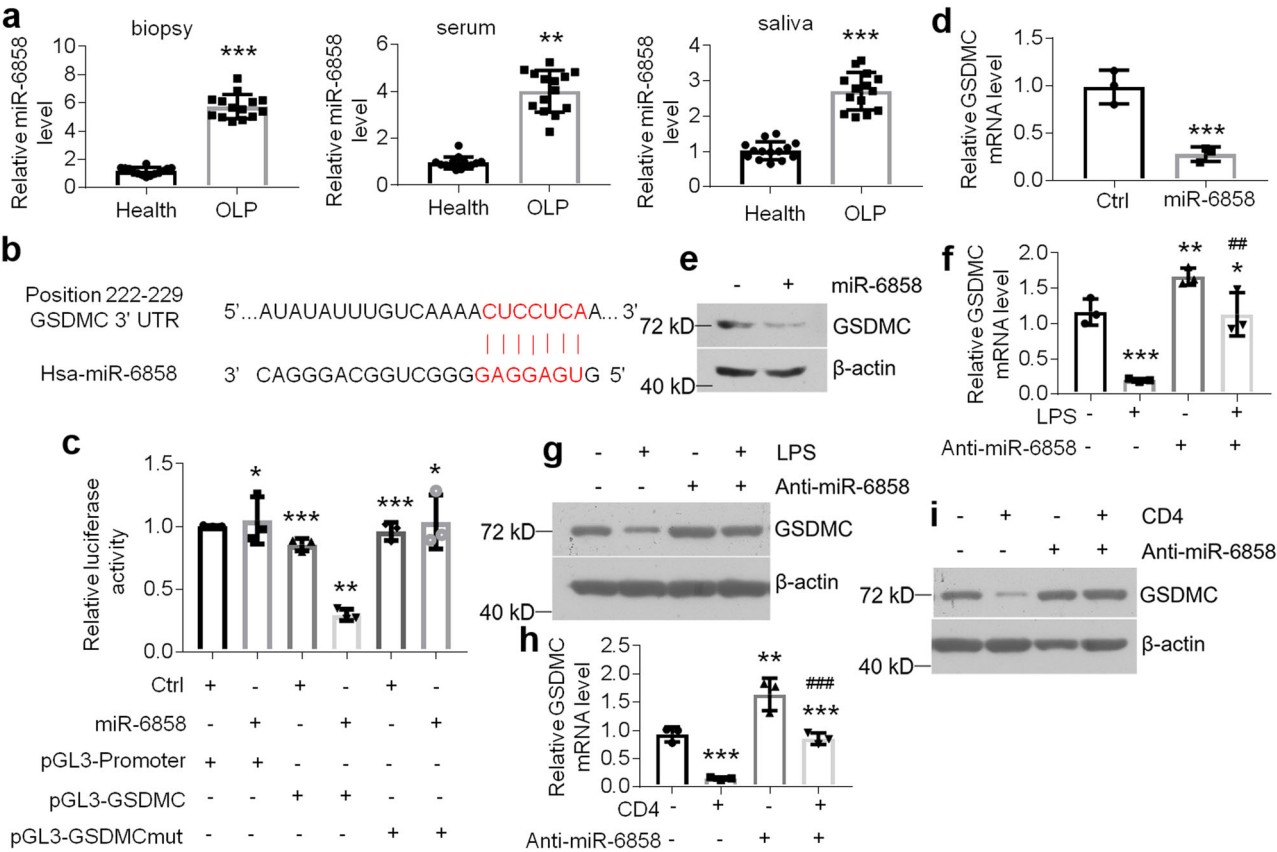

**Fig. 5 miR-6858 targets 3'UTR of *GSDMC* mRNA in human oral keratinocytes. a** m[6]A quantitation assays showing miR-6858 levels in oral keratinocytes of biopsies, serum and saliva derived from human individuals, $n = 14$. **b** Schematic showing the predicted miR-6858 target site in the 3'UTR of human *GSDMC* mRNA. **c** Luciferase activities of HOKs transfected with control or miR-6858 mimics, pGL3-Promoter, pGL3-GSDMC or pGL3-GSDMCmut plasmids as indicated, $n = 3$. GSDMC levels in HOKs with or without miR-6858 mimics evaluated by real-time PCR (**d**) or western blot (**e**), $n = 3$. Real-time PCR (**f**) or western blot (**g**) analyses showing GSDMC levels in control or miR-6858 inhibitors-transfected HOKs in the presence or absence of LPS treatment, $n = 3$. Real-time PCR (**h**) or western blot (**i**) analyses showing GSDMC expression in control or miR-6858 inhibitors-transfected HOKs in the presence or absence of activated CD4[+] T cells treatment, $n = 3$. The concentrations of microRNA oligonucleotides or plasmids are 200 nM or 500 ng, respectively. **$P < 0.01$, ***$P < 0.001$ vs corresponding control or health group; #$P < 0.05$, ##$P < 0.01$, ###$P < 0.001$ vs corresponding LPS or CD4 group; Ctrl, control. Data were expressed as means ± standard deviation. All experiments were carried out at least 3 times. Student's $t$ test (**a**, **d**) and one-way ANOVA (**c**, **f**, **h**) were used for statistical analysis.

apoptosis, consistent with other studies noting that *GSDMC* knockout is associated with apoptosis in tumor cell[25]. Moreover, *GSDMC* knockdown exhibits the ability to reduce colony formation in DLD-1 and LoVo cell lines[37], indicating GSDMC deficiency impedes cell proliferation in colon cancer. Mechanistically, the loss of GSDMC in colonic epithelial cells disrupted mitochondrial membrane potential and triggered cytochrome c release to activate the pro-apoptotic pathway[38]. Therefore, GSDMC may regulate cell apoptosis in oral keratinocytes through affecting mitochondrial homeostasis. We hypothesize that GSDMC may influence the stability of apoptosis-related factors such as Bax and Bak, and this hypothesis needs to be confirmed by more extra data. However, overexpression of GSDMC is reported to initiate pyroptosis in intestine[39], confirming the conclusion that GSDMC switches apoptosis to pyroptosis in cancer cells.

*GsdmC* is suggested to be directly fine-tuned by RNA m[6]A methylation in mouse colonic epithelial cells. On the contrary, we found that *GSDMC* is indirectly mediated by RNA m[6]A methylation in human oral keratinocytes. This discrepancy might be due to the differences on cell types or species. The reports concerning miR-6858 are rare, only one paper suggests that miR-6858 regulates the process of melatonin inhibition in

glioma[40]. Herein, we indicate that GSDMC decreases in oral keratinocytes is due to the upregulation of miR-6858 levels in the context of OLP. miR-6858 targets the 3'UTR of *GSDMC* mRNA to promote its degradation, in alignment with other investigations demonstrating miRNAs play critical roles in OLP[27,31]. Furthermore, consistent with other studies claiming m[6]A modification regulates primary microRNA processing and alternative splicing[34], we elucidated that elevated METTL14 in oral keratinocytes binds to m[6]A site of primiR-6858 to facilitate its processing, leading to miR-6858 upregulation. We also used METTL14 or METTL3 mutant to confirm that the regulation of primiR-6858 is dependent on m[6]A modification since the roles of METTL14 or METTL3 are not only limited in RNA methylation[35].

However, overexpression of GSDMC or inhibition of miR-6858 can not completely rescue cell apoptosis in OLP cell models because other factors or pathways may be involved in the process of cell death. In contrast to intestinal tissues, overexpression of GSDMC fails to trigger pyroptosis in oral keratinocytes, implying the distinct functions of GSDMC in different tissues.

In this exploration, we identified a factor GSDMC which is related to oral keratinocyte apoptosis in the setting of OLP.

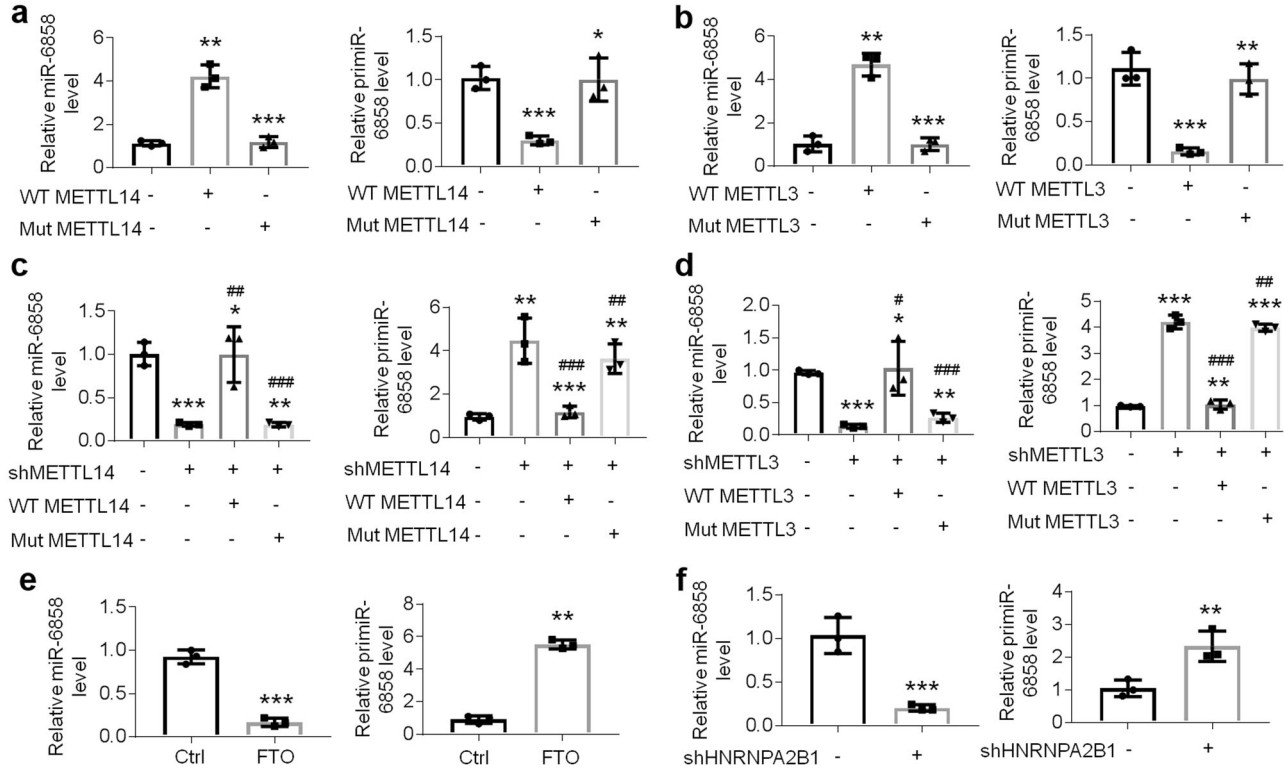

**Fig. 6 m6A modification regulates primiR-6858 processing.** Real-time PCR showing primiR-6858 or miR-6858 expression in HOKs infected with wild type/ mutant METTL14- (**a**) or METTL3- (**b**) lentivirus. **c** Real-time PCR showing primiR-6858 or miR-6858 expression in HOKs with shMETTL14-, wild type METTL14- or mutant METTL14-lentivirus as shown. **d** Real-time PCR analyses of primiR-6858 or miR-6858 expression in HOKs infected with shMETTL3-, wild type METTL3- or mutant METTL3-lentivirus as shown. **e, f** Primary miR-6858 or miR-6858 levels in HOKs with FTO- or shHNRNPA2B1-lentivirus determined by real-time PCR. $n = 3$. *$P < 0.05$, **$P < 0.01$, ***$P < 0.001$ vs corresponding control group; #$P < 0.05$, ##$P < 0.01$, ###$P < 0.001$ vs corresponding shMETTL14 or shMETTL3 group. Data were expressed as means ± standard deviation. All experiments were carried out at least 3 times. Student's *t* test (**e** and **f**) and one-way ANOVA (**a**, **b**, **c** and **d**) were used for statistical analysis.

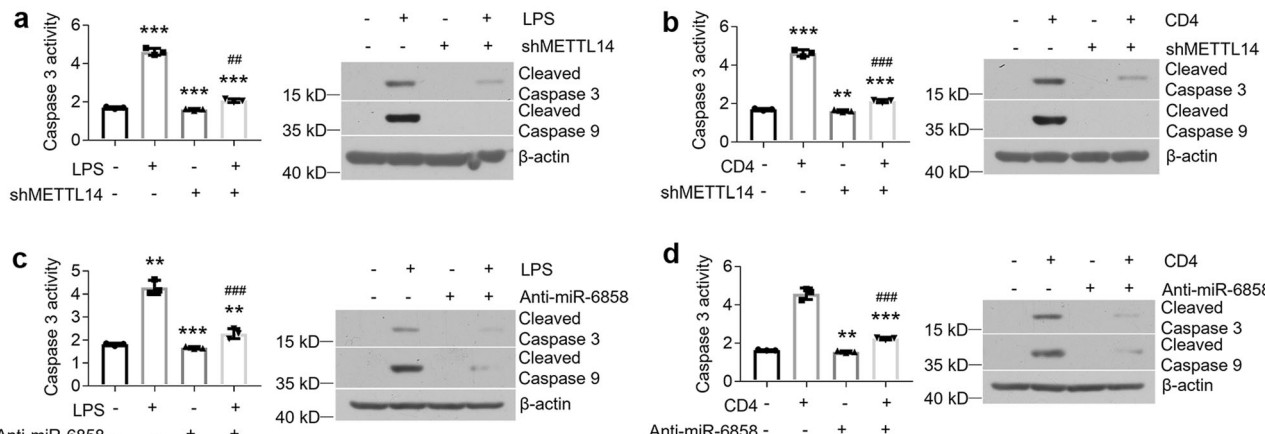

**Fig. 7 METTL14 or miR-6858 suppression blocks LPS- or activated CD4+ T cells-triggered cell apoptosis.** Caspase 3 activity and protein expression in control- or shMETTL14-lentivirus-infected HOKs with 12-hour LPS (**a**) or activated CD4+ T cells (**b**) treatment. Caspase 3 activity and protein expression in 200 nM control- or miR-6858 inhibitors-transfected HOKs with 12-hour LPS (**c**) or activated CD4+ T cells (**d**) treatment. $n = 3$. **$P < 0.01$, ***$P < 0.001$ vs corresponding control group; ##$P < 0.01$, ###$P < 0.001$ vs corresponding LPS or CD4 group. Data were expressed as means ± standard deviation. All experiments were carried out at least 3 times. One-way ANOVA (**a**, **b**, **c** and **d**) was used for statistical analysis.

Mechanistically, m6A methylation has a regulatory effect on primiR6858 to decrease GSDMC expression but not on GSDMC itself (Fig. 8f). Although we are unable to carry out experiments in vivo due to the lack of animal models mimicking OLP so far, the human and HOKs data we provided here are compelling to help us make conclusions. Given RNA methylation's roles in cell apoptosis, FTO inhibitors might be helpful for OLP management.

## Methods

**Human samples.** Human oral mucosa, blood and saliva were obtained from healthy and OLP participants in the Hospital of Stomatology of Shanxi Medical University[31]. OLP inclusion and exclusion criteria were according to the modified World Health Organization (WHO) diagnostic rules. This project was approved by the Institutional Ethical Committee of Shanxi Medical

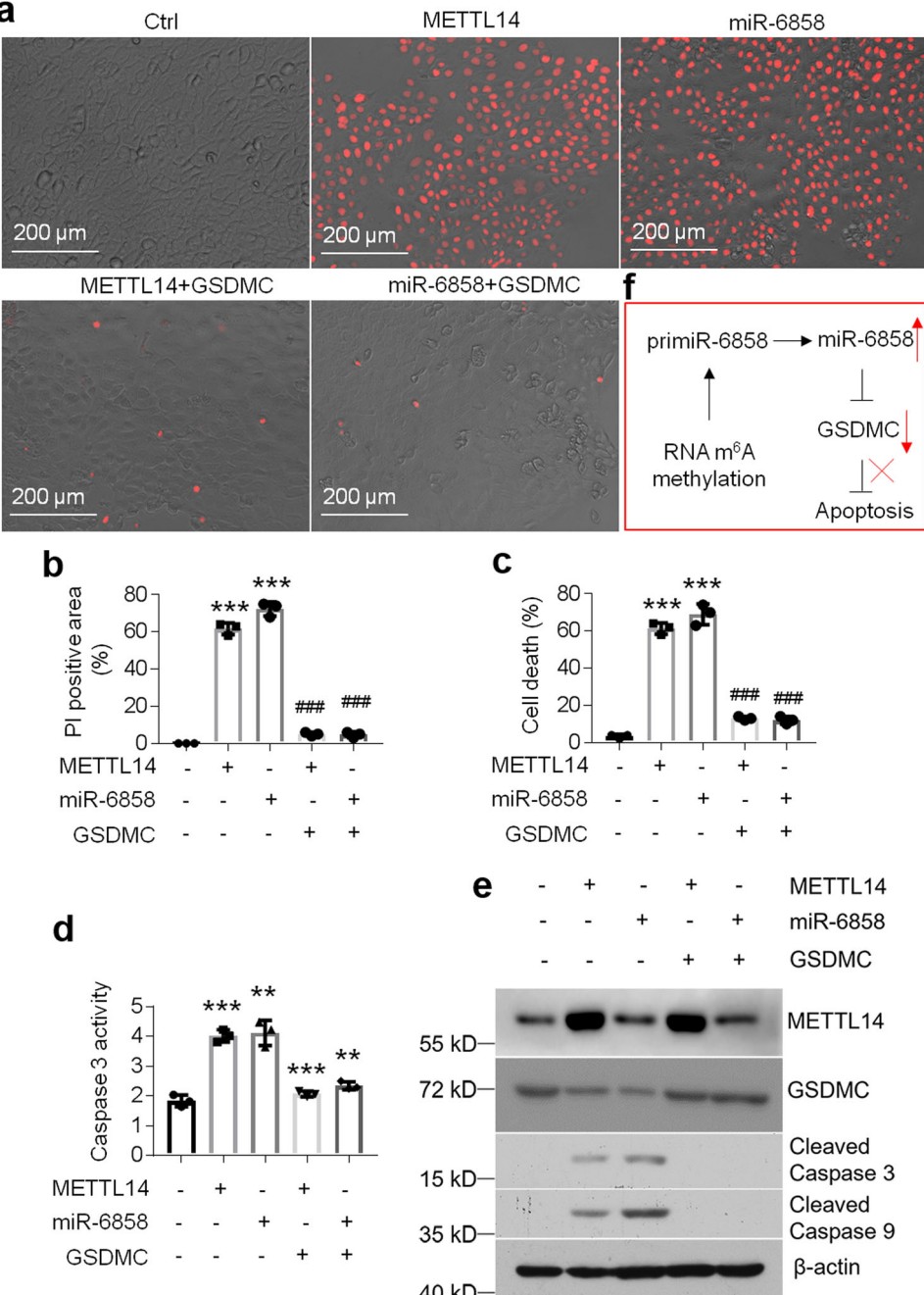

**Fig. 8 GSDMC overexpression rescues METTL14- or miR-6858 overexpression-induced cell apoptosis.** PI staining (**a**) and analysis (**b**), cell death measurement (**c**), caspase 3 activity assay (**d**) or western blot (**e**) in HOKs with METTL14-expressing lentivirus or miR-6858 mimics in the presence or absence of GSDMC-expressing lentivirus infection. **f** Schematic illustration of the m6A methylation-miR-6858-GSDMC axis in oral keratinocyte. The concentration of microRNA oligonucleotides is 200 nM. $n = 3$, **$P < 0.01$, ***$P < 0.001$ vs corresponding control group; ###$P < 0.001$ vs corresponding METTL14 or miR-6858 group. Data were expressed as means ± standard deviation. All experiments were carried out at least 3 times. One-way ANOVA was used for statistical analysis.

University (# 2016LL046). Informed consent from each individual was obtained and all ethical regulations relevant to human research participants were followed. Details of OLP patients' information was listed in previous studies[31].

**Cell culture**. HOKs were obtained from Chinese Shanghai Institution and cultured in plates using oral keratinocyte medium supplemented with fetal bovine serum (10%) and penicillin/streptomycin (1%). OLP is identified to be mainly triggered by T lymphocytes, the main lymphocytes involved are CD4+ and

CD8 + T cells[41]. In addition, bacterial infection is considered to be a prominent contributing factor to OLP[27]. To this end, we used two methods to resemble OLP in vitro. First, lipopolysaccharides (LPS, 100 ng/ml, Sigma-Aldrich, St. Louis, MO) was added into the culture medium of HOKs for 12 h. Second, the supernatants from anti-CD3/28-activated CD4+ T cells were added into the culture medium of HOKs for 12 h at a 30% final volumetric concentration[27]. Isolation and activation of T cells from human were performed according to a protocol[27]. In brief, peripheral blood samples were collected and CD4 + T cells were

enriched by anti-human CD4 magnetic particles (BD Biosciences, Franklin Lakes, NJ, USA). Anti-human CD3 and anti-human CD28 antibodies (BD Biosciences Pharmingen, SanDiego, CA, USA) were selected to activate CD4 + T cells. In another experiment, HOKs were challenged for 0, 4, 8, 12, hours by LPS or activated CD4$^+$ T cells, respectively. For miR-6858 mimics or inhibitors (mybiosource, San Diego, CA) treatment, HOKs were transfected with 200 nM oligonucleotides for 36 h.

**Oral mucosal epithelium isolation**. Oral mucosal epithelium was separated from buccal tissues which were digested in 0.25% dispase II for 12 h by muscle forceps as reported[27].

**Western blotting**. Proteins from human biopsies or HOKs were isolated using laemmli buffer. The same amounts of protein samples were separated by SDS–PAGE and subsequently transferred onto a polyvinylidene fluoride (PVDF) membrane (Millipore, Burlington, MA). After being blocked by 5% non-fat milk, membranes were treated with first and secondary antibodies. The information of primary antibodies was listed in Supplementary Table 1.

**RT-PCR**. Total RNAs from human tissues or HOKs were harvested using TRIzol (Invitrogen, Waltham, MA). The PrimeScript RT Reagent Kit (TaKaRa, Dalian, China) was used to synthesize the first strand cDNAs. The SYBR Premix Ex Kit (TaKaRa) was selected to perform qPCR in a real-time PCR system (Roche480). For miRNAs analyses, the miRNA isolation Kit (QIAGEN, Hilden, Germany) was chosen to extract miRNAs from human samples or cells. The specific miRNA First-strand cDNA Synthesis Kit (Aidlab Biotechnologies, Beijing, China) and a miRNA real-time PCR Assay Kit (Aidlab Biotechnologies) were applied to carry out the first strand cDNAs synthesis and qPCR, respectively. The relative amounts of transcripts were calculated using the $2^{-\Delta\Delta Ct}$ formula. GAPDH or U6 were regarded as internal control for mRNA or miRNA examinations. The equal amount of exogenous cel-miR-39 was served as internal control for circulating miRNAs in human serum and saliva. The primers for qPCR were listed in Supplementary Table 2.

**Lentiviral and plasmid constructs**. The coding sequence of human *METTL14*, *METTL3*, *FTO, GSDMC*-FL, cDNA was amplified and subcloned into the pLV[Exp]-Neo-EF1A lentiviral vector (VectorBuilder, Chicago, IL). Site-specific mutation of METTL14 or METTL3 was generated using a QuickChange Site-Directed Mutagenesis Kit (Agilent, Santa Clara, CA)[35]. For luciferase reporter plasmids construction, DNA sequence (50 bp) containing the potential target site of miR-6858 in the 3′UTR of human *GSDMC* cDNA was synthesized and inserted in pGL3-Promoter vector (Promega, Madison, WI). The inserted sequence is: GAGTTTATATATTTGTCAAAACTCCTCAAATAGTATGT TAAAGACGTAAG. pGL3-GSDMCmut was constructed by mutating the target sequence 5′CUCCUCAA3′ to 5′CUGGGCA A3′ in pGL3-GSDMC plasmid by a Mutagenesis Kit. The METTL14, METTTL3 and HNRNPA2B1 shRNAs were obtained from Sigma and co-transfected with lentivirus packaging plasmids to generate lentivirus in HEK293T cells. A shRNA for *HNRNPA2B1* (TRCN0000001058), a shRNA specific targeting *METTL3* (TRCN0000034715), a shRNA specific for *METTL14* (TRCN0000015933) and a control shRNA (SHC002) were used for experiments. The knockdown efficiencies of purchased shRNAs were all validated. Plasmids carrying GSDMC shRNA sequence were subcloned in pLV[shRNA]-EGFP-U6 lentiviral backbone (VectorBuilder)[37]. Scrambled shRNAs were served as controls for gene knockdown experiments (CCTAAGGTTA

AGTCGCCCTCGCTCGAGCGAGGGCGACTTAACCTTAGG). Empty vectors were used as controls for gene overexpression experiments. Negative miRNA mimic or inhibitor controls (Mybiosource) were selected for miRNA assays. All cloned DNA fragments and mutations were confirmed by DNA sequencing.

**Transfection and transduction assays**. MicroRNA oligonucleotides (200 nM) and plasmids were transfected transiently into HOK by Lipofectamin 3000 Reagent (Invitrogen). For transduction, lentivirus particles were added into HOKs at 10 MOI supplemented with 6 μg/ml polybrene for 36-hour culture.

**Luciferase reporter assay**. Luciferase reporter assays were carried out according to the protocol mentioned before with minor modifications[18]. In brief, HOKs plated onto 48-well plates were co-transfected with 200 nM miR-6858 mimics or 500 ng pGL3-Promoter, pGL3-GSDMC or pGL3-GSDMCmut plasmids by Lipofectamine 3000 reagent. After 36-hour incubation, HOKs were lysed and luciferase activities were measured using the Dual Luciferase Reporter Assay System (Promega) in a Lumet LB 9507 luminometer (Berthold Technologies, Bad Wildbad, Germany) and normalized to the activities of Renilla.

**Cell death assays**. Cell death was detected using the CytoTox 96 Non-Radioactive Cytotoxicity Assay kit (Promega, Catalog #G1780) according to the manufactures' instruction.

**m⁶A quantitation**. The m$^6$A quantitation assays in total cellular RNAs were performed using an EpiQuik m$^6$A RNA Methylation kit (Epigentek, Farmingdale, NY) in terms of the manufacturer's instruction. Briefly, total RNAs extracted from cells were mixed with capture antibody in detection wells at 37 °C for 90 min. After several times washes, the absorbance of solutions was detected at 450 nm using a microplate reader.

**m⁶A RNA-immunoprecipitation (RIP)-qPCR**. The Magna MeRIP m$^6$A Kit (Millipore) was used to perform m$^6$A RIP assay. Briefly, total RNAs were fragmented by RNA Fragmentation Reagents. 5% samples were saved as input controls. The rest fragmented RNA samples were mixed with magnetic beads conjugated with m$^6$A antibody for precipitation. The enriched RNA fragments were reversed-transcribed into cDNA and then quantified by real-time PCR. The enrichment of m$^6$A was normalized to the input controls. Primers were listed in Supplementary Table 2.

**Cross-linking and RNA immunoprecipitation (CLIP)**. CLIP assays were performed according to previous studies[18]. HOKs were crosslinked by UV light at 0.15 J/cm$^2$ and lysed in lysis buffer (150 mM KCl, 50 mM HEPES-KOH (pH 7.5), 1 mM NaF, 2 mM EDTA-NaOH (PH 8.0), 0.5% NP-40, 0.5 mM DTT, 1 μl/ml RNase inhibitor and 1x protease inhibitor cocktail) for nuclear extraction. After 15 min 1 U/ml RNase T1 treatment at 22 °C, 5% lysates were saved as input controls and the remaining lysates were rotated with control or anti-HNRNPA2B1 antibody-conjugated protein G magnetic beads for 1 h in cold room. Then the mixture was enriched with a magnet and washed by an IP wash buffer and a high-salt buffer, followed by complete resuspension in a proteinase K buffer. IP and input RNAs were extracted by TRIzol reagent and subjected to RT-qPCR. Primers for CLIP assays were listed in Supplementary Table 2.

**RNA affinity chromatography**. RNA affinity chromatography assay was carried out according to previous investigation[16]. Biotin-labelled ssRNA oligonucleotides harboring adenosine or

$m^6A$ were from GE Dharmacon (Lafayette, CO). After being denatured at 99 °C for 10 min, the ssRNA baits were put on ice immediately. 0.4 pmol RNA oligonucleotide was mixed with 50 μl streptavidin magnetic beads (Thermo Fisher Scientific) in binding buffer (200 mM NaCl, 20 mM Tris, 5 mM potassium fluoride, 5 mM β-glycerophosphate, 6 mM EDTA, 2 μg/ml aprotinin at pH 7.5) for 4 h in cold room. The RNA bait-beads complexes were then incubated with 200 μg nuclear extract derived from HOKs in binding buffer in a final volume of 400 μl in cold room for 12 h. After a set of washes, RNA–protein mixtures were dissolved in laemmli buffer and subjected to western blot analysis.

**Caspase 3 activity**. Caspase-3 activities in oral keratinocytes were assessed according to a protocol[27]. In brief, $1 \times 10^6$ oral keratinocytes were dissolved in lysis buffer for 10 min. After 10-min centrifugation at $3000 \times g$, supernatants were harvested as cell lysate for analysis. Caspase-3 activities in lysate were evaluated by caspase substrate Ac-DEVD-AFC (Bio Vision, Milpitas, CA), and monitored using a plate reader under Ex360/Em530 condition. Data of caspase 3 activity were normalized to cell number of HOKs.

**Image acquisition of cell death**. Cell death in HOKs was determined by propidium iodide (1 μg/ml) staining and images were captured using a fluorescent microscope. Quantitative analysis was performed using Image J software.

**ChIP assays**. ChIP assays were performed according to a published protocol with some modifications[42]. Briefly, oral keratinocytes were fixed with formaldehyde (1%) and neutralized by glycine. The chromatins in cell lysates were then sheared to get 400–500 bp fragments by sonication. 10% sheared samples were collected as input controls. The remaining 90% lysates were purified and mixed with anti-p65 or control IgG antibodies (4 μg). After O/N incubation, mixture was precipitated by protein A agarose beads. Following elution and purification with a set of washes, the precipitated samples were quantified by qPCR. The primers for qPCR were listed in Supplementary Table 2.

**CRISPR/Cas9-mediated knockout of Caspase3, Caspase7, GSDMD, MLKL, BAX and BAK**. sgRNA sequences targeting the Caspase3 gene (5′- ATGTCGATGCAGCAAACCTC-3′), Caspase7 gene (5′- CGTTTGTACCGTCCCTCTTC-3′), GSDMD gene (5′- CGGCTCTCACCTGTCGCGGG-3′), MLKL gene (5′-TTGAAGCATATTATCACCCT-3′), BAX gene (5′- TTTCTGACGGCAACTTCAAC-3′) and BAK gene (5′- GCATGAAGTCGACCACGAAG-3′) were subcloned into lentiCRISPRv2 vector (Addgene, catalog 52961). Packaging plasmids and lenti-vector were co-transfected into HEK293T cells. Lentivirus was collected from the cell culture medium after 48 h and infected into HOKs with 4 μg/ml polybrenes.

**Mitochondrial membrane potential measurement**. Mitochondrial membrane potential was detected by flow cytometry with the MitoProbe JC-1 Assay Kit (ThermoFisher, Cat#:M34152) according to the manufacturers' instruction using a BD LSRFortessa system (BD Biosciences).

**Mitochondria isolation**. Cytosolic and mitochondrial fractions were separated and harvested by a Mitochondria Isolation Kit (ThermoFisher, Cat#:89874) according to the manufacturer's instruction.

**Statistics and reproducibility**. All experiments were carried out at least 3 times independently. The biological replicates of human samples or HOK experiments are $n = 14$ or $n = 3$, respectively. Data were expressed as means ± standard deviation. Student's $t$ test was performed for two groups statistical analysis and ANOVA was used for analyzing the significant difference among three or more than three groups. $P < 0.05$ was identified to be significant.

**Reporting summary**. Further information on research design is available in the Nature Portfolio Reporting Summary linked to this article.

## Data availability

The data and materials are available from the corresponding author upon request or from supplemental materials. Source data or supplementary data files are provided with this paper (Supplementary Data 1–5). The uncropped blots were provided as in supplementary figures (Supplementary Fig. 7).

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

## Acknowledgements

This study was supported by National Natural Science Foundation of China grants (81800499, 81960565, 81560275), Fourth National Oral Health Epidemiology Survey Follow-up Research Project (201502002), Fundamental Research Program of Shanxi Province (202103021224231), Research Project of Shanxi Provincial Health Commission (2023049), Hainan Provincial Key Research and Development Program (ZDYF2020147), Hainan Natural Science Foundation Innovation Research Team Project (2018CXTD350) and a grant from Hainan Province Clinical Medical Center. We thank Jie Du (Shanxi Medical University) for his assistant work for this project.

## Author contributions

X.G. conceived and designed the research. X.W., S.L. ST, Y.D., R.G., H.S. and X.S. performed the experiments. R.L. collected human samples. X.W. analyzed data. X.G. and wrote the manuscript. X.W. and Y.D. acquired funding.

## Competing interests

The authors declare no competing interests.
