## [Peer Review File · Communications Biology]

Reviewer #1 (Remarks to the Author):

In this study, the authors attempted to elucidate the mechanism by which human oral keratinocytes (HOKs) undergo apoptosis during oral lichen planus (OLP). They found that OLP and its mimicry stimuli cause a decrease in GSDMC expression, which leads to apoptosis of HOKs. This study also suggests that OLP mimicking stimuli induce the up-regulation of METTL14 via NF- κ B, resulting in an increase in mirR-6858, which in turn decreases GSDMC expression.

This is an intriguing study demonstrating that inflammatory stimuli trigger apoptosis by counteracting the anti-apoptotic effect of GSDMC through RNA regulation. While a previous study has reported the anti-apoptotic effect of GSDMC (Du et al. *Dev Cell*, 2022), this concept is not yet well-established. Therefore, this study, which demonstrates independently consistent results, is timely and holds significant value. Experiments are well-controlled, and the results solidly support the authors' conclusions.

Major Comments

1. HOKs underwent apoptosis after knocking down GSDMC, but there are also cell types that do not express GSDMC without gene silencing (<https://www.proteinatlas.org/ENSG00000147697-GSDMC/single+cell+type>). This suggests that GSDMC is not necessarily required for apoptosis inhibition. The authors should provide a discussion on why the depletion of GSDMC immediately leads to apoptosis in HOKs.
2. In a previous report, it has been shown that METTL14 directly targets mouse *Gsdmc* mRNAs to increase its expression (Du et al. *Dev Cell*, 2022). This finding differs from the conclusions of the authors using human cells. It may be necessary to discuss the inter-species differences to understand this disparity.
3. The authors reached the conclusion that human GSDMC mRNA is not a direct target of METTL14, based on the predictive tool SRAMP database (lines 201-203). However, given that mouse *Gsdmc* mRNAs are directly targeted by METTL14, experimental evidence should be presented to substantiate this conclusion.
4. The experimental results in Fig. S7 are significant as they demonstrate the involvement of endogenous METTL14 and mirR-6858 in the induction of apoptosis by inflammatory stimuli. Therefore, these results should be included in the main figure. Additionally, the authors should present the levels of GSDMC in those Western blot samples (Fig. S7).
5. The authors' interest seems to be focused on the regulation of GSDMC expression. However, I am curious about the mechanism by which this protein controls apoptosis. Their experimental system could be valuable for elucidating this mechanism, such as identifying GSDMC binding partners or investigating which domains are essential for its anti-apoptotic activity.

Reviewer #2 (Remarks to the Author):

<<Summary:>>

The manuscript entitled "METTL14-upregulated miR-6858 triggers cell apoptosis in keratinocytes of oral lichen planus through decreasing GSDMC" by Xiangyu Wang and colleagues proposes a role for the methyltransferase METTL14 in destabilization of the mRNA transcript of GSDMC by miR-6858 resulting in increased cell death of oral keratinocytes augmenting inflammation. Specifically, the

authors suggest that activation of NFKB pathway leads to upregulation of METTL14 thereby increasing the levels of N6-adenosine methylation. Increased m6A methylation promotes processing of pri-miR-6858 and results in GSDMC mRNA degradation. This pathway is important in oral lichen planus a chronic inflammatory disorder in which patients exhibit increased inflammation in oral keratinocytes. The finding that reduction of GSDMC expression results in apoptotic cell death is line with a previous study showing GSDMC downregulation in the colon resulting in cell death. The interplay between microRNA and the GSDMC expression level is novel and interesting and can provide insight into oral keratinocyte cell death. Overall, the authors provide extensive work showing an interplay between methylation, mRNA stability and cell death, given that a previous study proposed a different function for METTL14 in reducing GSDMC the authors will need to address the difference to the manuscript.

<<Major:>>

- Why is the mechanism so different to the colon? Specifically, the study by Du Jie et al., *Developmental Cell*, 2022 (<https://doi.org/10.1016/j.devcel.2022.07.006>) shows that m6A methylation of GSDMC by Mettl14 is required for mucosal epithelial homeostasis in the colon. In contrast to the data in this manuscript Du Jie et al., found that absence of Mettl14 results in reduced GSDMC protein expression driving apoptotic cell death, impaired colon mucosal morphogenesis, increased mucosal permeability and inflammation. While in both studies reduction of GSDMC results in cell death the signalling pathway leading to the decrease of GSDMC is opposed. Do the authors have any explanation why m6A methylation has opposing roles in the colon compared to oral keratinocytes?

- o Along this line why does TNF α not induce cell death in HCT116 cells in the study by Du Jie et al.?

Could the authors include data using HOK cells treated with TNF- α and show the impact it has on cell death?

- The authors provide an extensive array of different techniques and a great number of experiments to support their conclusion however the lack of description and explanation in the manuscript makes the study harder to read and understand. Therefore, the reader would benefit greatly from a revised, edited and clarified manuscript especially to ensure the readers follow the authors experiments as well as conclusion. Examples:

- o Lack of information in the figure legends, particular concentrations of drugs added, timing etc.
- o Grammar as well as sentence structure
- o Description of experiments and reasoning e.g. use of CD4

- The cell death/caspase-3 activity data in Fig. 2c/e and Fig. 3a-c differs greatly in that cell death in Fig. 2c shGSDMC measure at ca. 60% cell death corresponding to a 2 fold increase in caspase activity, however in Fig. 3b/c caspase activity is also doubled compared to control but cell death is only at 20%. Moreover, in the Supplementary Fig. 1b GSDMC is almost completely reduced suggesting it should have a high potential to induce cell death? Why do the authors think this is the case and NFKB activation is unable to induce more cell death even though levels of GSDMC are highly reduced?

<<General comments>>

- Please clarify experimental setup, conditions, concentrations and timepoints in the figure legends
- o Fig.1b/c: why are 3 bands are shown however in the figure legends n=14?
- o Is Fig. 1c the quantification of b? Similar question for e/f and h/i?
- o Fig. 2e: How was the caspase-3 activity measured what was it normalized to, please add the information in the Method description.
- o Fig. 3: How long and what concentration was CD4 and LPS added to the cells. This is particularly important in this figure as it will allow the author to estimate the amount of GSDMC.
- o Fig. 4: The title of the figure does not describe the data in the figure, I would suggest to change it to "...METTL14 levels in oral keratinocytes instead.

- Please review figure titles are not overstating the data shown in the figures
one example being: Figure title of Fig.3 "... OLP cell models" the authors do not provide a specific OLP cell model but rather the use of human oral keratinocytes – please change the figure titles where appropriate.

- The authors could make the manuscript more accessible by reducing panels in individual figures e.g. immunoblot quantification can be added to the supplement rather than in the main figures.

- It would be helpful for the reader to see immunoblots showing full length as well as the cleaved form/s of caspases. Please provide full blots.

<<Minor and figure specific comments:>>

- Fig. 1: Do the authors also see higher levels of other NFKB target genes such as IL-1b ,Tnf- α , CCL5 etc. supporting NFKB activation and downstream signalling?

- Fig. 3: Here the authors use LPS and observe a reduction in GSDMC and increased cell death, does TNF- α or other TLR2/TLR4 agonists or NFKB inducers also reduce GSDMC expression and increased cell death? Particularly, TNF- α has been used in the study

<https://doi.org/10.1016/j.devcel.2022.07.006> and has not augmented cell death – does it induce cell death in HOK cells?

- Fig. 4: I couldn't find the effect of METTL-14 protein upregulation on the level of GSDMC expression, have I missed it? As the authors suggest METTL-14 being responsible for methylation and downstream targeting of GSDMC for degradation downregulation of GSDMC by immunoblot/RT-PCR is essential.

- In Fig.5c: The authors use a luciferase assay to show miR-6858 targeting of GSDMC using mutated and wt GSDMC, however it is unclear what control for miR-6858 was used and whether the miR-6858 was actually targeting the mRNA. Would it be a possibility to look at transcripts using RT-PCR in addition?

- Fig. 5e: As Fig. 5e shows miR-6858 targeting of GSDMC it would be interesting to see the levels of METTL14 which should not be upregulated in response to miR-6858 or anti-miR-6858 this would confirm miR-6858 being downstream of METTL-14. Have the authors tested this?

- Fig. 6: The authors overexpress the demethylase FTO and silenced HNRNPA2B1 (reader of m6A modifications) and show that this results in miR-6858 increase and miR-6858 decrease. What effect does FTO overexpression and HNRNPA2B1 silencing have on GSDMC expression and cell death? Equally does FTO overexpression or HNRNPA2B1 silencing in cells wherein METTL-14 is overexpressed rescue cell death?

- Fig. 7e: It would be interesting and nice to for the reader to see levels of GSDMC and METTL-14 in this experiment in addition to caspase-3 and caspase-9.

- Finally, can the authors speculate why in healthy individuals LPS does not induce increased m6A methylation, miR6858 upregulation, METTL14 upregulation and reduction of GSDMC? What is different in these individuals? Also do the results suggest that any oral infection with bacteria will trigger massive apoptosis – and is the source of the inflammation?

Responses to reviewers' comments

Reviewer #1: In this study, the authors attempted to elucidate the mechanism by which human oral keratinocytes (HOKs) undergo apoptosis during oral lichen planus (OLP). They found that OLP and its mimicry stimuli cause a decrease in GSDMC expression, which leads to apoptosis of HOKs. This study also suggests that OLP mimicking stimuli induce the up-regulation of METTL14 via NF- κ B, resulting in an increase in mirR-6858, which in turn decreases GSDMC expression.

This is an intriguing study demonstrating that inflammatory stimuli trigger apoptosis by counteracting the anti-apoptotic effect of GSDMC through RNA regulation. While a previous study has reported the anti-apoptotic effect of GSDMC (Du et al. Dev Cell, 2022), this concept is not yet well-established. Therefore, this study, which demonstrates independently consistent results, is timely and holds significant value. Experiments are well-controlled, and the results solidly support the authors' conclusions.

Response: We thank Reviewer #1 for the constructive comments. The manuscript has been rewritten and we have carefully addressed these comments.

Major Comments

1. HOKs underwent apoptosis after knocking down GSDMC, but there are also cell types that do not express GSDMC without gene silencing (<https://www.proteinatlas.org/ENSG00000147697-GSDMC/single+cell+type>). This suggests that GSDMC is not necessarily required for apoptosis inhibition. The authors should provide a discussion on why the depletion of GSDMC immediately leads to apoptosis in HOKs.

Response: This is a good point. Yes, one gene has different functions in different cell types. For example, GSDMD is mainly reported to regulate pyroptosis in macrophages (PMID: 26375003, PMID: 27281216). However, recent studies show that GSDMD is able to regulate mucus layer formation and food tolerance in intestine (PMID: 35119941, PMID: 37327784). Although we can not provide evidence to confirm that GSDMC mediates cell death in other tissues or cell types, we think GSDMC regulates cell apoptosis in colonic and oral epithelial cells at least based on our and previous studies. As discussed in the previous publication (PMID: 35917813), we propose that GSDMC may lead to apoptosis in HOKs through regulating mitochondrial homeostasis. GSDMC may influence the stability of apoptosis-related factors such as Bax and Bak, and this hypothesis needs to be confirmed by more extra data. This potential mechanism has been discussed in the revised manuscript.

2. In a previous report, it has been shown that METTL14 directly targets mouse Gsdmc mRNAs to increase its expression (Du et al. Dev Cell, 2022). This finding differs from the conclusions of the authors using human cells. It may be necessary to discuss the inter-species differences to understand this disparity.

Response: Thank you for this good suggestion. The m6a sites are variable in different cell types or species. For example, the m6a sites of Socs1 mRNA in mouse T cells and macrophages are largely different (PMID: 28792938, PMID: 33220174). GsdmC is suggested to be directly fine-tuned by RNA m6A methylation in mouse colonic epithelial cells. On the contrary, we found that GSDMC is indirectly mediated by RNA m6A methylation in human oral keratinocytes. This discrepancy might be due to the differences on cell types or species. This finding has been discussed in this revision.

3. The authors reached the conclusion that human GSDMC mRNA is not a direct target of METTL14, based on the predictive tool SRAMP database (lines 201-203). However, given that mouse Gsdmc mRNAs are directly targeted by METTL14, experimental evidence should be presented to substantiate this conclusion.

Response: This is a really good point. We mined the published database (GSE213714) from a previous publication (PMID: 36249071). Experimental evidence has been provided in this revision (Fig. S4 k-l).

4. The experimental results in Fig. S7 are significant as they demonstrate the involvement of endogenous METTL14 and mirR-6858 in the induction of apoptosis by inflammatory stimuli. Therefore, these results should be included in the main figure. Additionally, the authors should present the levels of GSDMC in those Western blot samples (Fig. S7).

Response: Thank you. These problems have been addressed according to your good suggestions. The levels of GSDMC in Fig. S7c-d (now Fig. 7c-d) had been provided in Fig. 5g-i, others are provided as follows:

5. The authors' interest seems to be focused on the regulation of GSDMC

expression. However, I am curious about the mechanism by which this protein controls apoptosis. Their experimental system could be valuable for elucidating this mechanism, such as identifying GSDMC binding partners or investigating which domains are essential for its anti-apoptotic activity.

Response: This is a very good point. Having said in the revision, GSDMC may regulate cell apoptosis in oral keratinocytes through affecting mitochondrial homeostasis. We hypothesize that GSDMC may influence the stability or movement of apoptosis-related factors such as Bax and Bak. We will carry out more extra experiments to confirm this hypothesis in our next project.

Reviewer #2 (Remarks to the Author):

<<Summary:>>

The manuscript entitled "METTL14-upregulated miR-6858 triggers cell apoptosis in keratinocytes of oral lichen planus through decreasing GSDMC" by Xiangyu Wang and colleagues proposes a role for the methyltransferase METTL14 in destabilization of the mRNA transcript of GSDMC by miR-6858 resulting in increased cell death of oral keratinocytes augmenting inflammation. Specifically, the authors suggest that activation of NF κ B pathway leads to upregulation of METTL14 thereby increasing the levels of N6-adenosine methylation. Increased m6A methylation promotes processing of pri-miR-6858 and results in GSDMC mRNA degradation. This pathway is important in oral lichen planus a chronic inflammatory disorder in which patients exhibit increased inflammation in oral keratinocytes. The finding that reduction of GSDMC expression results in apoptotic cell death is in line with a previous study showing GSDMC downregulation in the colon resulting in cell death. The interplay between microRNA and the GSDMC expression level is novel and interesting and can provide insight into oral keratinocyte cell death. Overall, the authors provide extensive work showing an interplay between methylation, mRNA stability and cell death, given that a previous study proposed a different function for METTL14 in reducing GSDMC the authors will need to address the difference to the manuscript.

Response: We thank Reviewer #2 for the constructive comments. The manuscript has been rewritten and we have carefully addressed these comments.

<<Major:>>

- Why is the mechanism so different to the colon? Specifically, the study by Du Jie et al., *Developmental Cell*, 2022 (<https://doi.org/10.1016/j.devcel.2022.07.006>) shows that m6A methylation of GSDMC by Mettl14 is required for mucosal epithelial homeostasis in the colon. In contrast to the data in this manuscript Du Jie et al., found that absence of Mettl14 results in reduced GSDMC protein expression driving apoptotic cell death, impaired colon mucosal morphogenesis, increased mucosal permeability and inflammation. While in both studies reduction of GSDMC results in cell death the signalling pathway leading to the decrease of GSDMC is opposed. Do the authors have any explanation why m6A methylation has opposing roles in the colon compared to oral keratinocytes?

Response: This is a really good question. We mined the published database (GSE213714) from a previous publication (PMID: 36249071). Experimental evidence has been provided in this revision (Fig. S4 k-l). These data showed that GSDMC mRNA levels are not directly regulated by m⁶A methylation in

HOKs. The m6a sites are variable in different cell types or species. For example, the m6a sites of Socs1 mRNA in mouse T cells and macrophages are largely different (PMID: 28792938, PMID: 33220174). As said in this revision, GsdmC is suggested to be directly fine-tuned by RNA m6A methylation in mouse colonic epithelial cells. On the contrary, we found that GSDMC is indirectly mediated by RNA m6A methylation in human oral keratinocytes. This discrepancy might be due to the differences on cell types or species.

o Along this line why does TNF α not induce cell death in HCT116 cells in the study by Du Jie et al.? Could the authors include data using HOK cells treated with TNF- α and show the impact it has on cell death?

Response: Thank you. TNF α is able to induce cell death in primary cells derived from healthy tissues, but I think it is very hard to make cancer cells dead upon TNF α treatment alone. We treated HOKs with or without 100 ng/ml TNF α for 12 hours and then detected cell death. Interestingly, we found that TNF α treatment can trigger cell death in HOKs (As shown below). We noticed TNF α failed to trigger cell death in HCT116 cells, this may be due to that HCT116 is a kind of colon cancer cell. Here are the data:

- The authors provide an extensive array of different techniques and a great number of experiments to support their conclusion however the lack of description and explanation in the manuscript makes the study harder to read and understand. Therefore, the reader would benefit greatly from a revised, edited and clarified manuscript especially to ensure the readers follow the authors experiments as well as conclusion. Examples:

- o Lack of information in the figure legends, particular concentrations of drugs added, timing etc.
- o Grammar as well as sentence structure
- o Description of experiments and reasoning e.g. use of CD4

Response: Thank you for your good suggestions. These concerns have been addressed in this revision. The description of the use of CD4 has been added

in the Methods.

- The cell death/caspase-3 activity data in Fig. 2c/e and Fig. 3a-c differs greatly in that cell death in Fig. 2c shGSDMC measure at ca. 60% cell death corresponding to a 2 fold increase in caspase activity, however in Fig. 3b/c caspase activity is also doubled compared to control but cell death is only at 20%. Moreover, in the Supplementary Fig. 1b GSDMC is almost completely reduced suggesting it should have a high potential to induce cell death? Why do the authors think this is the case and NFkB activation is unable to induce more cell death even though levels of GSDMC are highly reduced?

Response: This is a good question. I think there are two reasons. First, the levels of GSDMC in HOKs upon LPS or CD4+ cell treatment (Fig. S1b) are still higher compared to those in HOKs with shGSDMC-lentivirus (Fig. 2f). Second, these assays related to cell death and caspase 3 activity were performed in different batches, which leads to this discrepancy. However, the trends of these assays are the same since dead cells in these assays all increased, suggesting this discrepancy has no effect on the conclusion of this study.

<<General comments>>

- Please clarify experimental setup, conditions, concentrations and timepoints in the figure legends

o Fig. 1b/c: why are 3 bands are shown however in the figure legends n=14?

o Is Fig. 1c the quantification of b? Similar question for e/f and h/i?

o Fig. 2e: How was the caspase-3 activity measured what was it normalized to, please add the information in the Method description.

o Fig. 3: How long and what concentration was CD4 and LPS added to the cells. This is particularly important in this figure as it will allow the author to estimate the amount of GSDMC.

o Fig. 4: The title of the figure does not describe the data in the figure, I would suggest to change it to "...METTL14 levels in oral keratinocytes instead.

Response: Thank you. These concerns have been addressed in this revision.

o For Fig. 1b/c, the 3 bands are representative of n=14 samples since there are only 10 wells in each SDS-PAGE gel.

o For Fig. 1c, the bands of western blot are representative of quantification data, which has been clearly defined in this revision.

o For Fig. 2e, all of the assays for caspase-3 activity were performed by using 1×10^6 HOKs. The data were normalized to cell number. This description has been added.

o For Fig. 3, the timing and concentration of treatments have been added. More details were described in the Methods.

o For Fig. 4, this title has been changed.

- Please review figure titles are not overstating the data shown in the figures one example being: Figure title of Fig.3 "... OLP cell models" the authors do not provide a specific OLP cell model but rather the use of human oral keratinocytes – please change the figure titles where appropriate.

Response: Thank you for this good suggestion. This concern has been addressed.

- The authors could make the manuscript more accessible by reducing panels in individual figures e.g. immunoblot quantification can be added to the supplement rather than in the main figures.

Response: Thank you. Some immunoblot quantification panels have been moved to supplemental figure 5. For the better layout of figures, we did not change the rest of them. We hope this reviewer will agree with us.

- It would be helpful for the reader to see immunoblots showing full length as well as the cleaved form/s of caspases. Please provide full blots.

Response: This is a good point. However, the full length is not available since the antibodies here are specific to detect the cleaved bands of caspase. We hope this reviewer will agree with us. Here is the information of these antibodies:

Anti-cleaved caspase 3 (Cell Signaling Technology Cat#: 9664S, RRID:AB_2070042)

Anti-cleaved caspase 9 (Cell Signaling Technology Cat#: 9505, RRID:AB_2290727)

<<Minor and figure specific comments:>>

- Fig. 1: Do the authors also see higher levels of other NFkB target genes such as IL-1b ,Tnf- α , CCL5 etc. supporting NFkB activation and downstream signalling?

Response: Thank you for this good point. We detected these genes by real-time PCR and found their levels were highly increased in the epithelial layer of OLP compared to controls. Here are the data:

- Fig. 3: Here the authors use LPS and observe a reduction in GSDMC and increased cell death, does TNF-a or other TLR2/TLR4 agonists or NFKB inducers also reduce GSDMC expression and increased cell death? Particularly, TNF-a has been used in the study <https://doi.org/10.1016/j.devcel.2022.07.006> and has not augmented cell death – does it induce cell death in HOK cells?

Response: This is a good point. We treated HOKs with or without 100 ng/ml TNFα for 12 hours and then detected GSDMC mRNA levels and cell death. Interestingly, we found that TNFα treatment can reduce GSDMC expression and trigger cell death in HOKs. We noticed TNFα failed to trigger cell death in HCT116 cells, this may be due to that HCT116 is a kind of colon cancer cell. Here are the data:

- Fig. 4: I couldn't find the effect of METTL-14 protein upregulation on the level of GSDMC expression, have I missed it? As the authors suggest METTL-14 being responsible for methylation and downstream targeting of GSDMC for degradation downregulation of GSDMC by immunoblot/RT-PCR is essential.

Response: Thank you. We detected GSDMC mRNA levels by real-time PCR in HOKs with control or METTL14 overexpression. We found the forced expression of METTL14 decreased GSDMC expression in HOKs. Since there

are a lot of data in the manuscript and the responses to reviewers will also be published, we decide to show these data here. Here are the data:

- In Fig.5c: The authors use a luciferase assay to show miR-6858 targeting of GSDMC using mutated and wt GSDMC, however it is unclear what control for miR-6858 was used and whether the miR-6858 was actually targeting the mRNA. Would it be a possibility to look at transcripts using RT-PCR in addition?

Response: Thank you. Negative miRNA mimic or inhibitor controls (Mybiosource) were selected for miRNA assays as described in Methods. The control here is validated random sequences and has been tested on mammalian cells and tissues without identifiable effects on known miRNA function. In Fig 5d, we showed the mRNA levels of GSDMC in HOKs with or without miR-6858 treatment by RT-PCR.

- Fig. 5e: As Fig. 5e shows miR-6858 targeting of GSDMC it would be interesting to see the levels of METTL14 which should not be upregulated in response to miR-6858 or anti-miR-6858 this would confirm miR-6858 being downstream of METTL-14. Have the authors tested this?

Response: Thank you. We transfected HOKs with control or miR6858 mimics and found miR6858 has no effect on METTL14 mRNA levels. Here are the data:

- Fig. 6: The authors overexpress the demethylase FTO and silenced HNRNPA2B1 (reader of m6A modifications) and show that this results in miR-6858 increase and miR-6858 decrease. What effect does FTO overexpression and HNRNPA2B1 silencing have on GSDMC expression and cell death? Equally does FTO overexpression or HNRNPA2B1 silencing in cells wherein METTL-14 is overexpressed rescue cell death?

Response: Thank you. Upon FTO overexpression or HNRNPA2B1 silencing, GSDMC expression was increased and cell death was decreased in HOKs as shown below. Moreover, FTO overexpression or HNRNPA2B1 silencing rescued cell death in HOKs overexpressing METTL14. Here are the data:

- Fig. 7e: It would be interesting and nice to for the reader to see levels of GSDMC and METTL-14 in this experiment in addition to caspase-3 and caspase-9.

Response: Thank you. Levels of GSDMC and METTL14 have been shown in this revision.

- Finally, can the authors speculate why in healthy individuals LPS does not induce increased m6A methylation, miR6858 upregulation, METTL14

upregulation and reduction of GSDMC? What is different in these individuals? Also do the results suggest that any oral infection with bacteria will trigger massive apoptosis – and is the source of the inflammation?

Response: This question is very interesting. In the oral cavity of a healthy individual, I think the bacterial homeostasis is controlled under normal conditions. However, in the setting of OLP, the uncontrolled bacteria may induce keratinocyte death. Microbial dysbiosis is closely associated with the pathogenesis of OLP (PMID: 36338092). Compared with the healthy group, *Neisseria*, *Haemophilus*, *Fusobacterium*, *Porphyromonas*, *Rothia*, *Actinomyces*, and *Campylobacter* are significantly increased in the OLP group (PMID: 36338092). Therefore, we speculate that these increased bacteria may contribute to the initiation of massive apoptosis and inflammation.

REVIEWERS' COMMENTS:

Reviewer #1 (Remarks to the Author):

The authors have addressed all comments appropriately. I now believe this manuscript is worthy of acceptance.

Reviewer #2 (Remarks to the Author):

The authors have addressed all my concerns, addressed my comments, and have made modifications where suitable. I thus do not have any further concerns regarding manuscript publication.